# Risk Scenario Evaluation for Intelligent Ships by Mapping Hierarchical Holographic Modeling into Risk Filtering, Ranking and Management

**Wenjun Zhang [1], Yingjun Zhang [1,\*] and Weiliang Qiao [2]**

[1]  Navigation College, Dalian Maritime University, Dalian 116026, China; zhangwenjun@dlmu.edu.cn
[2]  Marine Engineering College, Dalian Maritime University, Dalian 116026, China; xiaoqiao_fang@dlmu.edu.cn
\*   Correspondence: zhangyj@dlmu.edu.cn

**Abstract:** To identify and screen the risk scenarios for the navigation risk of intelligent ships, the analysis and evaluation of navigational risks were performed in this study. Risk scenarios were developed and evaluated by mapping the hierarchical holographic modeling (HHM) into risk filtering, ranking and management (RFRM). In detail, considering the insignificant influences of some factors on navigational activities, risk factors were filtered and ranked using the RFRM model. Seven final factors were successfully determined, including traffic flow, navigation environment understanding, ship–shore interaction capabilities, target recognition capabilities, communication equipment reliabilities, professional skills, and situation judgments. The results indicated that cargo security can be guaranteed by following navigational risk identification and screening steps, and thus our findings provide theoretical guidance for the dynamic management of maritime organizations and ship companies. In addition, the proposed methodology is desirable for making predictions on maritime traffic risks.

**Keywords:** intelligent ships; risk identification; risk scenarios; HHM-RFRM

## 1. Introduction

The rapid development of artificial intelligence in the maritime industry has promoted the probability of operating ocean-going intelligent ships. According to documents issued by the Maritime Safety Committee (MSC) affiliated with the International Maritime Organization (IMO) [1–3], it can be reasonably speculated that intelligent ships would play an important role in the sustainable development of the maritime industry. Obviously, the capacity for the autonomous sailing of intelligent ships has the advantages of high efficiency, energy saving and security. However, the safety issues associated with intelligent ships challenge their application, which has to be addressed for the sustainable development of artificial intelligence in the maritime industry. As early as 2006, "e-Navigation" was presented by the IMO, indicated as the birth of intelligent ships [4]. Later, in 2007, the new generation of ships, named unmanned surface vessels (USVs), made their first appearance on the 98th MSC [5]. However, real-time information interactions between shore-based stations and USVs present a serious challenge. When the sailing distance is beyond the influence of a navigational communication system, the power of the USVs would inevitably be lost. Subsequently, in 2018, the USV was further redefined as the Maritime Autonomous Surface Ship (MASS) on the 99th MSC, able to sail autonomously and receive/send information from/to stations [6]. According to the automation levels of an operating system, the controlled performance of ships is divided into four levels. On the first level, the ship operating system is directly controlled by crews, and only a small part of the systems can run automatically. On the second level, the main system of the ship can be operated automatically or controlled remotely by crews, while the failure diagnosis depends on manual operations; as a result, some extra crews are necessary to guarantee

navigational safety. On the third level, related operators are able to keep away from the ship and make strategic decisions remotely. The desire of the final level is to remove all manual operations on board the ship, in which case, navigational activities are completely autonomic, and the attitude adjustments and decision-making of collision avoidance can be performed by the ship's intelligent system. The process of ship automation is essentially the process of intelligent development [7]. At present, the USV system is widely studied and applied, corresponding to the mentioned third and fourth intelligence levels. With this background, navigational risk identification and screening of intelligent ships have been researched in the present study.

Navigational safety is one of the important contents of intelligent ships' development and arouses great concerns from researchers [8]. Extensive studies associated with navigational risk identification and screening of intelligent ships are presented both theoretically and practically [9]. To be specific, the work from Wrobel et al. [10] showed that a navigational risk identification model can be established using human factor analysis and hypothesis analysis. According to more than 100 reports of ship accidents, the MASS is selected as the accident objective, and the accident conditions are simulated. Utne et al. [11] use system theory process analysis (STPA) to identify and analyze hazards, and the results show that risk identification based on the Bayesian belief network (BBN) is effective. Bolbot et al. [12] proposed a new network security evaluation method of a ship system, where the analysis procedure of the initial network risks was enriched greatly. The mentioned procedure supported the identification and evaluation of the network attack scenarios. These datasets can be used to identify and evaluate the network risks for intelligent ship navigation or ship propulsion systems of inland waterways. By utilizing VBPO-HSET techniques, Fan et al. [13] established a navigational risk model with high precision, and analyzed the ship safety via four navigation conditions and four risk types of the MASS case. By combining with the STPA and the SynSS, Zhou et al. [14] developed a new safety-comprehensiveness method (STPA-SynSS) to identify navigational risk, where decision-making can be employed to eliminate and decrease hazards, and dangerous factors can be tracked continuously and managed perfectly in a closed-loop. However, this cannot overcome the difficulties of data shortage for identifying the navigational risks of cargo ships. Data-mining technology and hypothesis analysis were employed by Yao [15] for the navigational risks of cargo ships based on traditional ship accident reports. The navigational risks' early warning is implemented by an intelligent evaluation model. Aiming at the problem related to navigational safety of intelligent ships, much research has been performed at the Dalian Maritime University and other institutions, including the key technologies of shore-based monitoring and warning [16–18], detection and recognition of maritime perils [19–21], and navigational risk evaluation for unmanned vehicles under complex sailing conditions [22–24]. Recent advances in intelligent ships have been obtained to provide an important guidance for the development of intelligent systems.

Compared with the traditional freight transportation, intelligent ships have more advantages, but the potential problems related to navigational safety cannot be ignored. Recent studies associated with navigational risks of intelligent ships are mainly focused on fragmented aspects, such as optimal control for the navigational trace of intelligent ships [25], automatic route design [26], and risk decision [27]. Few studies have been implemented to evaluate the risks involved in the shipping operations for intelligent ships, which are essential for the industrial operation of intelligent ships. Therefore, in the present study, the risk scenarios involved in the shipping operations for intelligent ships are developed and evaluated at a macroscopic level. Firstly, hierarchical holographic modeling (HHM) and the risk filtering, ranking and management framework (RFRM) were combined to establish a navigational risk identification model for intelligent ships. Secondly, the navigational risk factors were identified and screened using the proposed model, and the key factors were obtained by filtering data. Finally, the results provide theoretical foundation for navigational safety, risk early warnings, and management measures of intelligent ships.

## 2. Principle for Mapping HHM into RFRM

### 2.1. HHM and RFRM

The HHM method was proposed by Haimes in 1981, and was regarded as a systematic and comprehensive methodology [28,29]. The purpose is to capture and demonstrate internal features and essences from different aspects, perspectives, views, dimensions, and hierarchies of one system. For large-scale, multi-objective and multi-level risks, the HHM method can solve problems by using multi-view and omnidirectional studies, and also analyze the sub-scenario separated from the whole in different views. Around 1991, the risk ranking and filtering (RRF) method was developed by the Center for Risk Management of Engineering Systems (CRMES) [30]. On this basis, the RFRM model was further established to meet the requirements of practical applications. Eight stages of this model are stated as follows: scenario recognition, scenario preliminary filtration, double filtration standard, multi-standard evaluation, quantitative rating, risk management, evaluation for filtering scenario, and actual scenario feedback. Note that the foregoing five stages can perform risk identification and filtration, and the last three stages can achieve risk management, feedback and modification. The objectives of this study were to identify and filter the key risk factors for the navigational safety of intelligent ships corresponding to the foregoing five stages, which are listed as follows:

(1) Scenario recognition: The different risk scenes are described by generating the HHM model for ship systems;

(2) Scenario preliminary filtration: This stage is achieved by filtering the risk scenes in above stage. The decision-maker provides primary evaluation based on literature reading and expert experience;

(3) Double filtration standard: First, the irrelevant risks are defined qualitatively according to the results and possibilities from the stage 2; then, the irrelevant risks are filtered based on the risk filtration matrix; finally, the sorting operation of effective risks is performed to improve the accuracy in the next stage;

(4) Multi-standard evaluation: For each risk scene, the evaluation should simultaneously consider results and possibilities of risk, reducibility, robustness and redundancy [31]. This stage focuses on measuring the risk factors' ability of defeating the defense system, where the rest risk scenes can be evaluated reasonably;

(5) Quantitative rating: A quantitative risk matrix is employed to evaluate the risk defined in stage 4, and the evaluation rank of risk scenes is further presented to identify the key risk factors.

### 2.2. Analysis for Risk Scenario

In 1981, Kaplan and Garrick [32] proposed the concept of risk scene generation to explain the definition of three sets, including risk scenario, occurrence probability, and damage degree. In general, the risk set is a complex multivariate set which cannot be represented as a number or a vector. The risk $R_{risk}$ can be defined as:

$$R_{risk} = \{(S_i, P_i, X_i)\}_c \tag{1}$$

where $S_i$ is the risk scenario; $P_i$ is the occurrence possibility of risk; $X_i$ is the damage degree; the subscript $c$ is the risk set and represents a complete set containing all possible risk scenarios (or all important risk scenarios). On this basis, the system normal scenario $S_0$ is defined, which conforms to the actual plan. The risk scenarios $S_1$ are evolved from the normal scenarios and widely used in risk identification and evaluation.

The further development of risk scenarios provides basic platform for the risk view construction. The risk views have evolved in various ways depending on the specific system scene. There are four kinds of risk views in engineering: analysis view from the shore-based operator (P); analysis view from the ship (S); analysis view from the environment (E); analysis view from the management (M). These risk views have a significant effect on the influence degree, occurrence probability and damage degree of the ship risk. Assuming that

all systems' risk scenarios are finite, A is defined as an ordinal set of the risk set $\{S_i\}$. To explain conveniently, the constraint condition can be described as:

$$R_{risk} = \{(S_a, P_a, X_a)\}_c, \; a \in A \tag{2}$$

The risk views are constructed as:

$$\left.\begin{array}{l} R_P = \{S_i, P_i, X_i\} \wedge a_P \in A \\ R_S = \{S_i, P_i, X_i\} \wedge a_S \in A \\ R_E = \{S_i, P_i, X_i\} \wedge a_E \in A \\ R_M = \{S_i, P_i, X_i\} \wedge a_M \in A \end{array}\right\} R = \{S_i, P_i, X_i\}_c \tag{3}$$

In general, three properties of the risk set $\{S_i\}$ for the risk analysis are summarized as follows:

(1)　Completeness: The union of all parameters can be formulated as $\{a_P \cup a_S \cup a_E \cup a_M\} = A$;
(2)　Finiteness: The risk scenarios in navigation are finite;
(3)　Separability: For $i \neq j$, $a_i \cap a_j = \varnothing$.

To combine the risk scenarios and HHM model, each component involved in the HHM model would be regarded as a description of the safety scenario of $S_0$. The scenarios deviating the boundary of $S_0$ would be regarded as the risk scenarios. In addition, the risk scenario established by the principle of scenario construction has to satisfy the requirement of completeness on the basis of the risk factors' identification. The objectives of risk identification are specific, so a certain amount of the risk scenarios must be screened. When the risk identification is carried out under the HHM framework, different views are reflected from different scenarios, which is interpreted by the separability of the risk set $\{S_i\}$. Noting that multiple perspectives make the risk identification representative, as a result, risk scenarios are not required to be completely independent from each other.

## 3. HHM Modelling for Navigational Risk Identification

The HHM is applied to identify the risk sources involved in the navigational activities of intelligent ships. Practically, the operation of maritime shipping can be divided into four stages: sailing plan decision; berthing and unberthing; port entry and exit; and open water navigation. Before the risk identification, the risk evaluation system first needs to be developed.

### 3.1. Identification of Risk Aspects

The composition and logical relationships of the risk aspects mainly embody the permutation and association of each risk-influencing factor. A comprehensive and reasonable composition is useful to classify and sort each risk factor; as a result, the overall risk levels of shipping can be described objectively and comprehensively [33]. In the entire field of safety research, safety and accidents are always studied in opposition to each other. In general, there are many unsafe factors hiding behind the accidents' backgrounds, leading these accidents to seem safe [34]. However, it is common sense that the hidden risk factors still exist objectively. These risk factors have not yet been triggered to arouse enough attention. Therefore, in the specific research process, researchers consider the direct impact factors of accidents as indicators of ship navigation safety, and those hidden potential impact factors are also contained. Therefore, the structure of the risk aspects is illustrated as shown in Figure 1.

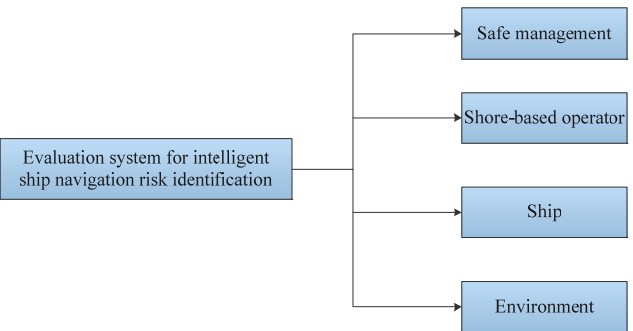

**Figure 1.** The distribution of risk aspects for intelligent ships.

According to the contents of Figure 1, it is noted that human, machine, environment and management are the four basic elements for safe navigation [35]. Based on the classification principles of these four elements, navigation risks are identified on the perspectives of safety management, the shore-based operator, the ship, and the external environment. By the aid of the aforementioned risk aspects, the hierarchical structure of these risk aspects is shown in Figure 2.

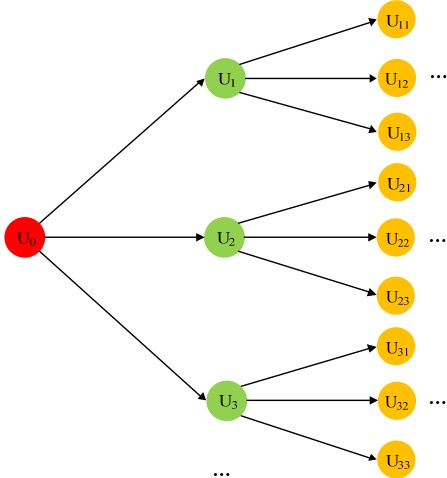

**Figure 2.** Illustration of the hierarchical structure for risk identification.

Different from traditional manned ships, intelligent ship risk identification is characterized by a certain degree of particularity. A lot of complicated factors need to be considered when selecting risk factors. However, there are no relatively definite criteria or standards when selecting these factors [36,37]. The selection of risk factors is mainly determined on the basis of the researcher's experiences according to their actual situation and requirements. Usually, it is necessary to select precise, feasible, and comparable risk factors, which are able to maximize the impact of these factors as much as possible. Subsequently, the maximum impact factor is considered as the main index to evaluate the overall risk levels of intelligent ships. In addition, the selected risk factors should have practical reference value, and can be compared with factors selected by other research methods, which can illustrate the rationality and feasibility of the proposed method. In general, two kinds of method were used in the present study to select risk factors: literature review and expert survey [38].

### 3.2. Analysis of Navigation Risk Factors

As mentioned above, the expert survey method is also referred to as the Delphi method [39]. When the risk evaluation index system is established, various risk impact

indexes are firstly selected and listed according to the evaluation objectives. Secondly, these indexes are made into questionnaires, and finally analyzed statistically by experts' comments. In addition, mathematical statistics can also be used to predict the expert recommendations of the survey, with the aim of calculating the risk index factors recommended by experts with a high degree of influence. In this study, we analyzed the navigation risk factors of intelligent ships by consulting different pieces of research and experts' opinions. Based on the iterative idea of the HHM process, a specific iterative analysis flow of risk factors is given, as shown in Figure 3. Moreover, two expert groups are employed in the present study, and the details for these experts are summarized in Table 1.

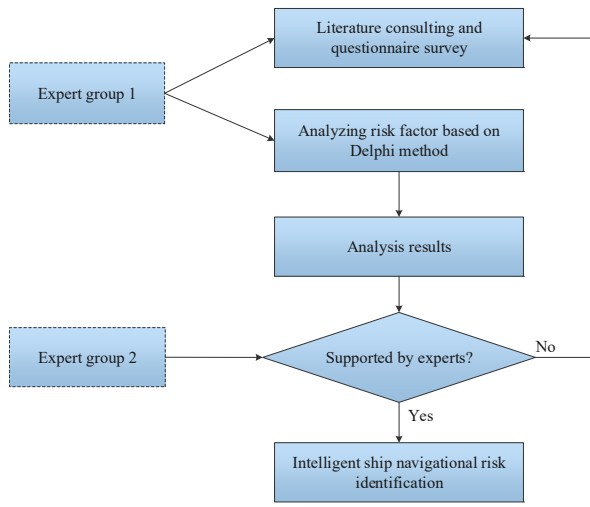

**Figure 3.** Flow chart of iterative analysis for risk indexes.

Expert group 1 included researchers from domestic and foreign maritime institutions, staff engaged in maritime management, and ocean shipping captains. Because these experts had more than ten years of work and management experience, and also had a deep understanding of navigation risk management for traditional ships and modern intelligent ships, the questionnaire survey was firstly conducted on them. The questionnaire included two parts. The first part listed the risk factors which were screened out after consulting the literature in the initial stage. The second part was designed with open questions, listing the risk factors of intelligent ship navigation based on experts' own experiences. Then, the risk source framework was enriched and improved. Finally, the risk factors of intelligent ship navigation were analyzed, and the preliminary analysis results were concluded with the Delphi method by organizing a group of experts. Expert group 2 included researchers, engineers, and professors from well-known maritime colleges, unmanned ship research institutes, ship design institutes, ship management institutes, etc. This group of experts audited the preliminary analysis results from the previous step and examined its comprehensiveness, feasibility and rationality. If the results were supported by the expert group 2, the HHM model could be developed on the basis of the analysis results. Otherwise, the improved suggestions would be given and the above procedures be repeated.

According to the principle of iterative analysis illustrated in Figure 3, three iterations were completed in this study, and relatively reasonable results based on the risk factor analysis were obtained. These risk factors were divided into four perspectives, according to the actual navigation condition of the intelligent ship, including human factors, ship factors, environmental factors, and management factors. Worthy of note is that human-related risk factors are produced by the shore-based operators of the remote-control center.

**Table 1.** General information for the employed experts.

| Expert | Age | Occupation | Educational Level | Certificate Rank | Job Tenure |
|--------|-----|------------|-------------------|------------------|-----------|
| Expert group 1 | | | | | |
| E1 | 50 | Maritime investigator | Master of navigation | Captain | He was employed as captain for 6 years before becoming a maritime investigator in maritime authority |
| E2 | 43 | Shipping manager | Master of navigation | Senior captain | He is currently in charge of a project addressing safety issues associated with intelligent ships |
| E3 | 46 | Seafarer | Bachelor of navigation | Senior captain | He is an experienced captain on an ocean-going ship which is characterized by high automation |
| Expert group 2 | | | | | |
| E4 | 53 | Professor | Doctor of navigation | Captain | He is a professor employed in a maritime university, and his research interests are focused on safety issues for intelligent ships |
| E5 | 45 | Senior engineer | Doctor of marine engineering | Chief engineer | He is employed by a high-tech institute aimed at the intelligent production of ships |
| E6 | 42 | Associate professor | Doctor of navigation | Captain | He has been focused on the safety issues of intelligent ships for more than 5 years |

### 3.2.1. Human Factors

At present, human factors still account for most of the various maritime accidents, such as the recent stranding incident of "Changci Vessel". In the field of risk evaluation research, the discipline of human factor engineering has emerged in recent years. The core of this discipline is to study the risks caused by human control factors [40]. In addition, Sotiralis et al. [41] proposed the theory of human error analysis, as well as error and missing information theory, in detail, suggesting that missing information leads to errors in decision-making, such as errors in judging the encounter distance and encounter time with other ships. There is also a violation of maritime traffic rules by human operation errors, such as failure to comply with maritime navigation rules or traffic separation rules while navigating at sea. In this respect, the impact of intelligent ships on navigation safety is much smaller than that of traditional ships' human factors. Because intelligent ships have introduced the concepts of remote controlling centers and shore-based operators, some human factors of traditional manned ships have been transferred to the shore-based operators. According to the definition and work restrictions of shore-based operators by the IMO [42], the influencing factors are analyzed from the overall perspective of intelligent ship's shore-based operators, including skill trainings, psychological status, physiological status, emergency responses, etc.

### 3.2.2. Ship Factors

For intelligent ships, the ship factors mainly include the ship's own influence, stowage, structure and performance, equipment and maintenance, perception and understanding, etc. The risk factors of the ship itself usually include: ship dimension, ship tonnage, and ship age [43]; the risks associated with ship stowage mainly refer to the full load rate of the cargo, the degree of cargo securing, and the dangerous property of the cargo. While sailing at sea,

the structure and performance, such as the ship speed, water tightness, and hull strength of the ship are closely related to its collision risk and loss. Under complex navigation conditions, especially in narrow waterways, shallow waters, bridge areas, high wind and wave conditions, and other extreme conditions, the perception and understanding of intelligent ships is particularly important. The perception system represents the traditional observation of the environmental information of manned ships. Therefore, when the intelligent ship is performing target recognition, various sensors are used to obtain multi-source heterogeneous information associated with the environment to understand the navigational environment, and then to make situational judgments, which are then entered into the intelligent ship navigation decision-making system through semantic descriptions, to make avoidance changes. The final goal of target recognition by making decisions and plans is to realize navigational safety. Perception and understanding mainly include target recognition ability, navigational environment understanding, situation judgment, and semantic understanding. During the sailing of an intelligent ship, it is necessary to maintain the contact and communication between the ship and the shore-based operation center. Therefore, the reliability of maritime communication equipment is the main factor that needs to be considered.

### 3.2.3. Environment Factors

Ocean environment factors have a great influence on the safe navigation of intelligent ships. Statistical data shows that a considerable number of maritime traffic accidents are caused by the complicated navigational conditions and/or harsh natural environments [44]. Due to the complexities of the marine environment, there are many uncertainties in hydrometeorological forecasts, traffic environment estimation, and other types of forecasts associated with time. These are difficult for humans to control and judge, and it may also be possible to erroneously judge the probability of accidents. In addition, in many cases, the harsh environmental situation is unavoidable. By comprehensively analyzing and studying various internal and external environmental factors, the environmental factors involved in intelligent ship risk evolution analysis can be divided into five aspects: meteorological conditions; hydrological conditions; navigational conditions; entry and exit conditions; and other interference factors. According to the demand analysis of intelligent ships for external navigation information, in the present study, the hydrometeorological indexes are specifically considered as visibility, wind, rain, thunder and lightning, illumination, flow rate and swell, etc. Simultaneously, the navigation conditions, entry and exit conditions and other interference factors are divided into traffic flow, obstruction, navigational aid condition, surplus water depth, berth utilization rate, port environment, floating debris of offshore, shipwreck and reef, etc. Note that some extra environmental factors affecting the navigational safety of the intelligent ships were not considered in this study.

### 3.2.4. Management Factors

Factors in the management of traditional manned ships mainly involve the management of the crew on board and the process management of teamwork, as well as maritime management departments and ship management companies. For intelligent ships, management factors mainly include corresponding ship management companies and maritime management departments. The focus of this study is mainly on the safe management system of intelligent ship companies. The factors are determined as shipping company's safe management systems, departmental cooperation, maritime supervisions, maintenance timeliness and maintenance cycles in maintenance plans, emergency management capabilities, technical support capabilities and ship–shore interaction capabilities. In the present study, the influencing factors of ship management for traditional manned ships and the characteristics of intelligent ship management were comprehensively considered to determine the multi-factors involved in management aspects.

*3.3. Development of HHM Model*

The navigational risk identification of intelligent ships is characterized by complexity, large-scale, and multi-level. The interactions among internal risk factors are complicated. The risk sources from different perspectives are generated on the interaction between the external environment and the ships. If the risk factors are only partially considered in the identification, the aim of controlling the risk effectively cannot be achieved. Therefore, it is extremely effective to analyze the risk factors in an all-round and multi-perspective manner by using an HHM model.

In order to select and reflect the risk sources comprehensively, the intelligent ship navigational risk identification with the HHM model was established based on risk factor analysis in this study. Firstly, four main scenarios were determined. Then, the main scenarios were decomposed into 20 levels of hierarchical holographic sub-scenarios. Finally, 20 hierarchical holographic sub-scenarios were decomposed into 38 sub-level holographic sub-scenarios. To sum up, there were 44 risk factors in total, decomposed from the main scenarios, which are illustrated in Figure 4.

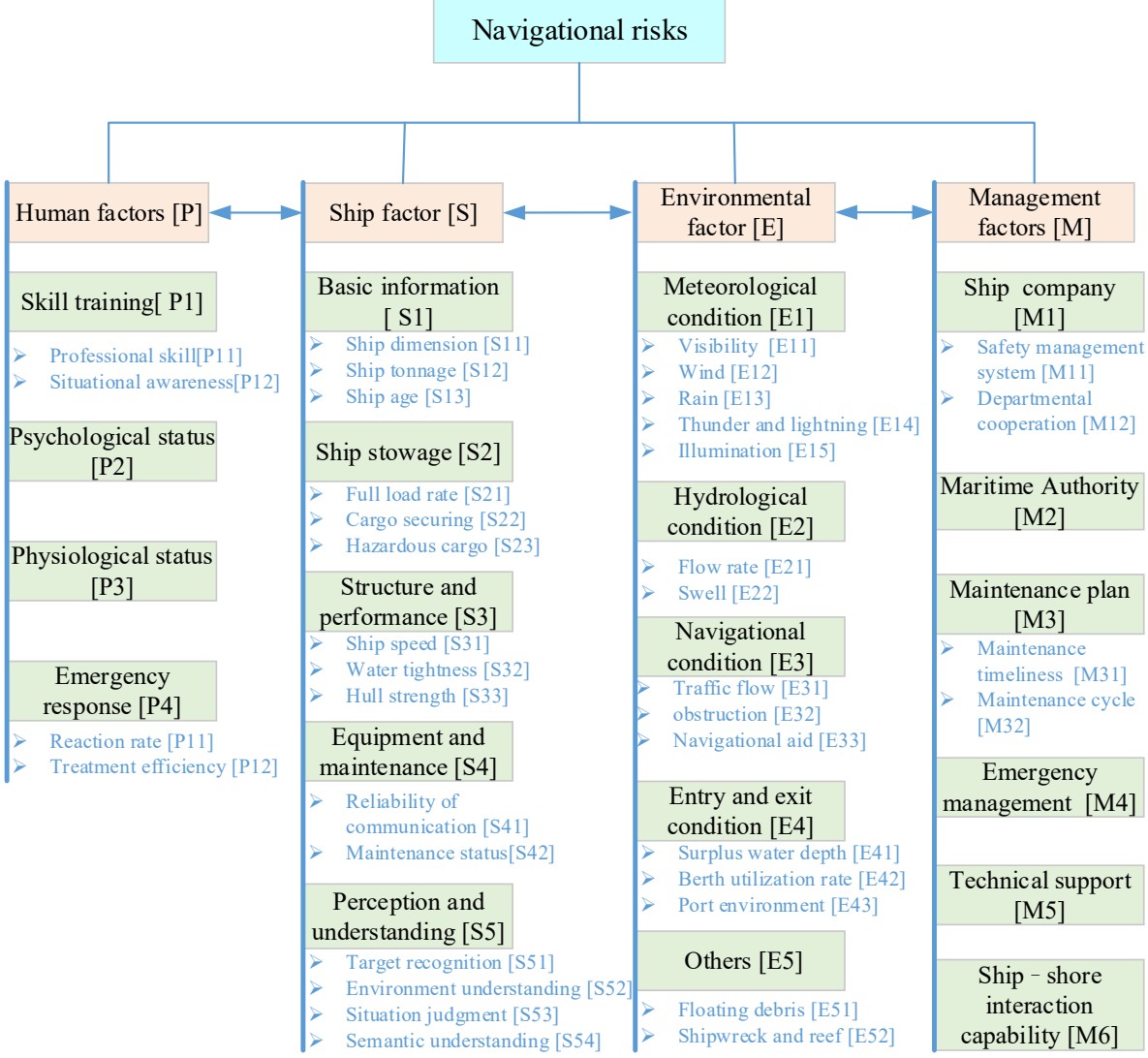

**Figure 4.** Framework of navigational risk identification for intelligent ships based on HHM model.

## 4. Results and Discussion

In accordance with the general process of RFRM and the characteristics of intelligent ship navigation, the identified risks were screened and evaluated, and the results are as follows.

### 4.1. Screening for the Identified Risks

In the first step, according to the HHM model established above, four main risk scenarios for the navigational safety of intelligent ships were identified, including shore-based operator analysis scenarios, ship analysis scenarios, environmental analysis scenarios, and management analysis scenarios.

The second step was to filter the main risk scenarios identified in the first step. The risk sources identified in the first step were relatively comprehensive, and many risk subsystems were constructed successfully. However, if each subsystem was refined, hundreds of risk sources may be generated. In fact, not all risk sources have a direct impact on intelligent ships' navigational risk. Therefore, the risk filtering was performed according to the risk scenarios that needed to be studied, and these scenarios were combined with the actual investigation results to reduce the number of risks for the next step. In this study, based on the survey opinions of 10 experts, 8 unimportant risk scenarios were screened out: physiological status (P3); ship age (S13); cargo dangerous property (S23); hull strength (S33); ship maintenance level (S42); lightning (E14); departmental cooperation (M12); and maintenance period (M32). As a result, there were 36 risk scenarios remaining.

The third step was to use the risk matrix to filter. The identification and filtering of risk scenarios cannot completely filter out the irrelevant risk scenarios. Therefore, it was necessary to combine the risk filter matrix to further filter out some insignificant risks. In order to improve the accuracy of the next analysis, the ranking of risks was performed preliminarily. There are two criteria for risk matrix filtering: one is the possibility of the risk occurrence, and the other is the influential consequence of the risk. The risk evaluation levels for the possibility of occurrence and influential consequence are shown in Tables 2 and 3, respectively.

**Table 2.** Evaluation levels for the possibility of single factor risk.

| Level | Possibility of Occurrence | Sign |
|---|---|---|
| High | Almost certain to happen | S |
| Higher | The probability of occurrence is very large | H |
| Medium | The probability of occurrence is medium | M |
| Lower | The probability of occurrence is small | L |
| Low | Almost impossible to happen | N |

**Table 3.** Evaluation levels for the influential consequence of single factor risk.

| Level | Influential Consequence | Sign |
|---|---|---|
| Serious | Cause a catastrophic impact, and it takes lots of manpower and resources to eliminate | S |
| Higher | Cause a greater impact, and it takes more manpower and material resources to eliminate | H |
| Medium | Cause a certain impact, and it takes a certain amount of manpower and resources to eliminate | M |
| Lower | Cause less impact, and less manpower and resources to eliminate | L |
| Neglected | Cause a little impact, and humans and resources can be ignored | N |

The severity of the risk is defined by combining the possibility of occurrence and influential consequence for the risk. According to the severity of the risk, it can be divided into four levels: high risk, relatively high risk, general risk and low risk. From the

perspective of decision makers, there is no need to spend too much time and resources addressing low risks, but high-risk and relatively high-risk events must be first considered. Therefore, when conducting risk management, the risk levels of low and general are generally filtered out, and the risk filtering and ranking matrix are shown in Table 4.

**Table 4.** Risk filtering and ranking matrix.

| Consequence | Possibility | | | | |
|---|---|---|---|---|---|
| | **Low** | **Lower** | **Medium** | **Higher** | **High** |
| Neglected | Low risk | Low risk | Low risk | Low risk | General risk |
| Lower | Low risk | Low risk | Low risk | General risk | Higher risk |
| Medium | Low risk | General risk | General risk | Higher risk | High risk |
| Higher | General risk | General risk | Higher risk | Higher risk | High risk |
| Serious | High risk | High risk | High risk | High risk | High risk |

In the second step, the risk scenarios, possibilities of risk occurrence and influential consequences of the risk were evaluated by the experts after their respective screenings. For the consequences of risks, the following rules were developed: if the consequences of risk were argued as serious by more than 6 out of 10 experts, then the consequences of the risk were extremely serious; if the consequences were evaluated by 6 experts to obtain a greater level, but did not reach a severity level which was regarded as a greater consequence, the risk was not as serious; the evaluation indexes for the medium and lower levels were similar, and other conditions could be ignored. The same rules were also applied to the possibilities of risks. Considering the suggestions from 10 experts on the consequences and possibilities of risks, a ranking on the possibilities of risk occurrence and influential consequences of the risk could be obtained. The risks filtered in the second step were put into the matrix, and the risks in the low-risk level and general-risk level were removed, as shown in Table 5.

**Table 5.** Intelligent ship navigational risk identification matrix.

| Consequence | Possibility | | | | |
|---|---|---|---|---|---|
| | **Low** | **Lower** | **Medium** | **Higher** | **High** |
| Neglected | | | S11 | M11, M4 | E12, E13, E15 |
| Lower | E42, E43 | | S12, S21 | | |
| Medium | | P12, P2, E32 | E51 | E21, E22 | |
| Higher | S32, E33 | P41, P42 | P11, M2, M5 | E11, E31 | S31 |
| Serious | E52, E41 | S22, S41 | | M31, M6 | S51, S52, S53, S54 |

| | Low risk | | General risk | | Higher risk | | High risk |
|---|---|---|---|---|---|---|---|

In the above risk identification matrix, after filtering out the low and general risks, a total of 18 risk factors were retained as important factors including professional skills, maritime supervision, technical support, visibility, ship speed, traffic flow, flow rate, swells, shipwreck and reefs, surplus water depth, cargo securing, reliability of communication, maintenance timeliness, ship–shore interaction, target recognition capabilities, environment understanding, situation judgment, and semantic understanding. However, the other 18 risk factors that were not retained in this study were also considered. Because the influences were not so high compared to the remaining 18 risk factors, they were regarded as the minor factors for the navigational safety of the intelligent ships and could be screened out. In order to simplify the research, we first analyzed the main risk factors for the navigational risk analysis of the intelligent ships. This is also compatible with the nature of the scenes set $\{S_i\}$ in the previous risk analysis. According to the contents of Table 5, the risk scenarios stemming from surplus water depth and shipwreck and reef were negligible based on perspective of probability; however, these two risk scenarios have to be paid attention due to the serious possible consequences, which have been found as the main causes for ship total loss according to Allianz [45]. In addition, it is interesting to find that the risk scenarios

associated with human factors were considered as less important for the intelligent ships than for regular ships, with similar results also obtained by Chang et al. [24], who argued that human errors are ranked as the third most important factors for the safety of intelligent ships. It was also noticeable that the technology-related factors represented by S51, S52, S53, and S54 were critical for the safe operating of intelligent ships based on Table 5, especially for the technologies associated with environmental perception and situation awareness, which are also paid considerable attention by Luo and Shin [46], Thieme and Utne [47], and Fan et al. [13].

### 4.2. Evaluation for the Screened Risks

The fourth step was multi-criteria evaluation. For the risk factors screened in the previous step, except for the two influencing factors of possibility of occurrence and influential consequence, the reducibility, redundancy and robustness of the system in each scenario should also be considered [31]. Worthy of mention is that the three characteristics are all risk elements that can defeat the system's defense. The documents involved in [48] are referred to for this study, and 11 criteria for the navigational risk evaluation of the intelligent ships are proposed, which are shown in Table 6.

**Table 6.** Multi-criteria evaluation for navigational risk of the intelligent ships.

| Number | Criteria | Description |
|---|---|---|
| $S_1$ | Unperceived | An existence mode of the initial event in a scene refers to the inability to discover before the accident |
| $S_2$ | Uncontrollability | No control method to adjust and avoid or prevent damage |
| $S_3$ | Various failure modes | Indicates that a certain factor has many ways to cause damage to the ship's navigation |
| $S_4$ | Irreversibility | Indicates that when a certain factor has a problem, it cannot be returned to the original normal state |
| $S_5$ | Duration of impact | The duration of the adverse consequences |
| $S_6$ | Cascading influence | Indicates that the influence of the impact factors of a certain subsystem can easily spread to other subsystems |
| $S_7$ | Operating environment | The sensitivity to unknown operating environment |
| $S_8$ | Loss | The loss of the ship's navigation system |
| $S_9$ | People/Organization | Scenes where adverse effects and consequences are amplified through the interface between multiple systems |
| $S_{10}$ | Complexity/Emergency | Indicates that a certain factor has system-level behavioral potential and has a certain degree of complexity |
| $S_{11}$ | Immaturity of design | The proved deficiency or absence of the system design |

The 11 risk scenarios listed in Table 6 are divided into three levels: "high (H)", "medium (M)" and "low (L)", where corresponding scores are shown in Table 6. Each classification can be described according to the multi-criteria evaluation for navigational risk of intelligent ships. In this manner, the risk scenarios filtered in the third step were evaluated successfully, and the non-significant risks were removed.

It can be seen from Table 7 that the importance of these 11 evaluation criteria was not exactly the same. Therefore, it was also necessary to determine the weight of each standard and apply the weight analysis method to calculate the weight of each standard, as shown in Table 8.

**Table 7.** Evaluation description for navigational safety of intelligent ships.

| Number (Criteria) | H (10 Points) | M (5 Points) | L (0 Points) |
|---|---|---|---|
| $S_1$ | Unperceived | Late unperceived | Early unperceived |
| $S_2$ | Not control | Difficult to control | Easy to control |
| $S_3$ | A lot of | A little | A single |
| $S_4$ | Irreversible | Partially reversible | Reversible |
| $S_5$ | Long | Medium | Short |
| $S_6$ | Many knock-on effects | Rarely contact | No contact |
| $S_7$ | Highly sensitive | Sensitive | Not sensitive |
| $S_8$ | A lot of | A little | Seldom |
| $S_9$ | Highly sensitive | Sensitive | Not sensitive |
| $S_{10}$ | Highly complex | Medium complex | Low complex |
| $S_{11}$ | Highly immature | Immature | Mature |

**Table 8.** Details of the weight for 11 evaluation criteria.

| Criteria $S_i$ | $S_1$ | $S_2$ | $S_3$ | $S_4$ | $S_5$ | $S_6$ | $S_7$ | $S_8$ | $S_9$ | $S_{10}$ | $S_{11}$ |
|---|---|---|---|---|---|---|---|---|---|---|---|
| Weight | 0.13 | 0.15 | 0.08 | 0.10 | 0.10 | 0.12 | 0.05 | 0.05 | 0.08 | 0.07 | 0.07 |

Through interviews with experts, evaluation results of 18 risk scenarios were obtained. The evaluation results of experts were directly converted into the corresponding scores, which were convenient for comparison and calculation. The results can be seen in Table 9.

**Table 9.** Results of risk evaluation.

| Risk ($R_i$) | Criteria ($S_i$) | | | | | | | | | | |
|---|---|---|---|---|---|---|---|---|---|---|---|
| | $S_1$ | $S_2$ | $S_3$ | $S_4$ | $S_5$ | $S_6$ | $S_7$ | $S_8$ | $S_9$ | $S_{10}$ | $S_{11}$ |
| P11 | 10 | 5 | 0 | 5 | 5 | 10 | 5 | 0 | 10 | 5 | 0 |
| S22 | 5 | 5 | 10 | 5 | 10 | 10 | 5 | 5 | 10 | 5 | 0 |
| S31 | 0 | 0 | 0 | 10 | 5 | 10 | 0 | 5 | 5 | 5 | 0 |
| S41 | 0 | 10 | 5 | 5 | 10 | 5 | 0 | 5 | 10 | 5 | 5 |
| S51 | 5 | 5 | 10 | 5 | 5 | 10 | 10 | 5 | 10 | 10 | 5 |
| S52 | 5 | 5 | 10 | 5 | 10 | 10 | 10 | 10 | 10 | 10 | 5 |
| S53 | 5 | 0 | 5 | 10 | 5 | 5 | 10 | 5 | 10 | 5 | 5 |
| S54 | 5 | 5 | 5 | 5 | 5 | 5 | 5 | 5 | 5 | 10 | 5 |
| E11 | 0 | 10 | 5 | 10 | 5 | 5 | 0 | 0 | 0 | 0 | 0 |
| E21 | 0 | 10 | 5 | 10 | 5 | 5 | 5 | 0 | 5 | 0 | 0 |
| E22 | 0 | 10 | 5 | 10 | 5 | 5 | 0 | 5 | 0 | 0 | 0 |
| E31 | 0 | 10 | 10 | 10 | 10 | 10 | 10 | 10 | 10 | 10 | 5 |
| E41 | 0 | 5 | 0 | 10 | 0 | 0 | 0 | 0 | 0 | 0 | 0 |
| E52 | 5 | 10 | 5 | 10 | 5 | 10 | 10 | 10 | 10 | 5 | 5 |
| M2 | 0 | 0 | 0 | 5 | 5 | 5 | 0 | 0 | 0 | 0 | 5 |
| M31 | 5 | 5 | 10 | 5 | 10 | 10 | 10 | 10 | 10 | 10 | 5 |
| M5 | 0 | 0 | 5 | 0 | 0 | 5 | 0 | 5 | 5 | 5 | 5 |
| M6 | 0 | 5 | 10 | 10 | 10 | 10 | 10 | 10 | 10 | 10 | 5 |

First, the score of each risk was multiplied by their corresponding weights. Then, the capacity scores of attacking the defense system for each scenario were obtained by combining these values. The capacity score function is described as:

$$R_i = \sum_{i=1}^{11} r_i w_i \tag{4}$$

For example, the risk of the professional skill (P11) is expressed as: $R_1 = 10 \times 0.13 + 5 \times 0.15 + 0 \times 0.08 + 5 \times 0.1 + 5 \times 0.1 + 10 \times 0.12 + 5 \times 0.05 + 0 \times 0.05 + 10 \times 0.08 + 5 \times 0.07 + 0 \times 0.07 = 5.6$. The final results are figured in Table 10.

**Table 10.** Results of capacity scores.

| Risk $R_i$ | P11 | S22 | S31 | S41 | S51 | S52 | S53 | S54 | E11 |
|---|---|---|---|---|---|---|---|---|---|
| Total score | 5.65 | 6.55 | 3.7 | 5.75 | 7 | 7.75 | 5.4 | 5.35 | 4 |
| Risk $R_i$ | E21 | E22 | E31 | E41 | E52 | M2 | M31 | M5 | M6 |
| Total score | 4.65 | 4.25 | 8.35 | 1.75 | 7.75 | 1.95 | 7.75 | 2.35 | 7.6 |

$R_i R_i$ The scores of the observations in Table 10 represent the risk capacity for attacking the defense system. The higher the scores are, the stronger the ability is, and vice versa. In this study, the risk of scoring less than 5 points was screened out, and the risk of scoring more than 5 points remained. On the basis of the third step, 7 risks were screened out, including ship speed (S31), visibility (E11), speed (E21), swells (E22), surplus water depth (E41), maritime supervision (M2) and technical support (M5), and 11 risks remained, needing to be investigated in further.

The fifth step was risk quantitative ranking. The quantization and ranking of the risks screened in the fourth step were carried out. The importance degree of the risk was determined by using the quantitative criteria. In this way, the decision-maker can avoid risk accurately, and the plan can be guaranteed to perform successfully. Tables 11 and 12 show the reference values of the possibility of occurrence and influential consequence of single-factor risks.

**Table 11.** Reference values of the possibility of occurrence of single-factor risks.

| Level | Quantitative Criteria | Sign |
|---|---|---|
| High | (80%, 100%) | S |
| Higher | (60%, 80%) | H |
| Medium | (40%, 60%) | M |
| Lower | (20%, 40%) | L |
| Low | (0%, 20%) | N |

**Table 12.** Reference values of the influential consequence of single factor risk.

| Level | Quantitative Criteria | Sign |
|---|---|---|
| Serious | (80%, 100%) | S |
| Higher | (60%, 80%) | H |
| Medium | (40%, 60%) | M |
| Lower | (20%, 40%) | L |
| Neglected | (0%, 20%) | N |

By using the reference values in Tables 11 and 12, the numerical criteria of the possibility of occurrence and the influential consequence was determined to evaluate the rank of

all risks. Table 13 shows the evaluation criteria. In this study, $R = p \cdot q$, where $R$ is the risk ranks of the single factor.

**Table 13.** Comprehensive evaluation criteria of single-factor ranks ($R$).

| Level | Quantitative Criteria | Influence Degree | Sign |
|---|---|---|---|
| Significant risk | (64%, 100%) | High possibility, large loss, the impact and loss are unacceptable | S |
| Higher risk | (36%, 64%) | Higher possibility, larger loss, the impact and loss are acceptable | H |
| General risk | (16%, 36%) | Little possibility, little loss, and generally does not affect the feasibility of the project | M |
| Lower risk | (4%, 16%) | Small possibility, small loss, and the feasibility of the project is not affected | L |
| Minimal risk | (0%, 4%) | Very small possibility, very small loss, and the impact on the project is small | N |

The concrete values of 11 risks screened from step 4 were calculated based on the Bayesian method [49]. Bayesian's equation is widely used in the field of natural science and maritime traffic safety, and can be written as [50]:

$$P(A|E) = \frac{P(A)P(E|A)}{P(E)} \tag{5}$$

$$P(E) = P(E|A)P(A) + P(E|\overline{A})P(\overline{A}) \tag{6}$$

In the present study, the prior probability and conditional probability required by Bayesian inference were obtained according to the judgment made by the six experts who were introduced in Table 1. Then, the post probability for each risk was calculated by Equations (5) and (6). As a result, the key risks were obtained, including traffic flow (E31), navigation environment understanding (S52), ship–shore interaction capabilities (M6), target recognition (S51), reliability of communication (S41), professional skills (P11) and situation judgment (S53). Details are listed in Table 14. Therefore, these seven risk scenarios should be paid much attention for the industrial operation of intelligent ships. For instance, the potential failure of ship–shore interaction represented by M6 is regarded as the most serious risk scenario, which is also supported by most scholars involved in intelligent ship study [13,51]. The influence of traffic flow may determine the operation mode of intelligent ships; for intelligent ships sailing in water of heavy traffic, the ships may have to be manned to avoid maritime accidents. According to the results presented in Table 14, the environmental understanding capacity denoted by S52 is critical for the safety of intelligent ships; for this purpose, many advanced technologies have been proposed to improve perception for intelligent ships, such as high-definition cameras [52], the multi-scale ship-detection approach [53] and the intelligent collision-avoidance technique [54]. In addition, even though the influence of human-related factors on the safety of intelligent ships is alleviated by adapting intelligent technologies, the human-related factor of professional skill denoted by P11 is still determined as one of key factors in this study. It should be noted that the professional skill hereinafter is associated with operators in land-based centers rather than the seafarers on board the ships. The risks stemming from the operators are also considered greatly by Prison et al. [55], Jalonen et al. [56] and Fan et al. [13]. Overall, the risk scenarios identified in Table 14 are essential for the sustainable development of intelligent ships, which may be valued for the future research of the safety issues of intelligent ships.

**Table 14.** Risk matrix for intelligent ships.

| Consequence | Possibility | | | | |
|---|---|---|---|---|---|
| | $0 \leq R \leq 4\%$ | $4\% < R \leq 16\%$ | $16\% < R \leq 36\%$ | $36\% < R \leq 64\%$ | $64\% < R \leq 100\%$ |
| Neglected | | | | | |
| Lower | | | | | |
| Medium | | | | | |
| Higher | | | E52, S22 | M31, S54 | |
| Serious | | | | S51, S41, P11, S53 | E31, S52, M6 |

| Low risk | General risk | Higher risk | High risk |
|---|---|---|---|

### 5. Conclusions

By the aid of the existing navigational risk research and case studies, this paper proposes an HHM-RFRM-based risk identification and screening methodology for intelligent ship navigation and, where feasible, scientific and reasonable factors were considered. The risk evaluation criteria system of intelligent ship navigation was constructed, including shore-based operators, ships, environment, and management. The risk factors were screened out, and seven factors were determined as the key factors, including traffic flow, navigation environment understanding, ship–shore interaction capabilities, target recognition capabilities, reliability of communication, professional skills, and situational judgment. By comparison with practical cases, this research shows excellent adaptability and reliability in risk management, thus providing a reference value. The results indicate that the proposed method can provide theoretical guidance for decision-makers to take risk measures and technology guidance for the safe navigation of intelligent ships. This is valuable for the sustainable development of intelligent ships.

**Author Contributions:** Conceptualization, Y.Z.; methodology and formal analysis, W.Z.; writing, W.Q. All authors have read and agreed to the published version of the manuscript.

**Funding:** This research was funded by the Program of Innovative Talents of Dalian (Grant number 2021RQ043) and LiaoNing Revitalization Talents Program (Grand number XLYC1902071).

**Institutional Review Board Statement:** Not applicable.

**Informed Consent Statement:** Not applicable.

**Data Availability Statement:** Not applicable.

**Conflicts of Interest:** The authors declare no conflict of interest.

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
