# Peer review of "Risk Scenario Evaluation for Intelligent Ships by Mapping Hierarchical Holographic Modeling into Risk Filtering, Ranking and Management"

_sustainability, doi:10.3390/su14042103_

Round 1

Reviewer 1 Report

*in the discussion section, the results obtained in the study should be discussed. It is useful to compare the obtained data with the literature or other studies

*Information about how experts were selected, and their qualifications were not provided in the study.

*Has the Bayesian model been used in the study? This situation is not understood.

*The journal publishes studies on sustainability. It is useful to emphasize the relationship of the study with sustainability. In particular, it can be related to this study in the introduction and conclusion sections.

Author Response

Dear Editors and Reviewers:

Thank you for your letter and for the reviewers’ comments concerning our manuscript entitled “Risk scenarios evaluation for intelligent ships by mapping hierarchical holographic modeling into Risk Filtering, Ranking and Management”. (ID: Sustainability-1591128). The authors are very appreciative of the review made by the reviewers and the valuable comments and recommendations provided for the improvement of the above-referenced manuscript. We have studied comments carefully and have made correction which we hope meet with approval. Revised portion are marked in red in the manuscript. The point to point responses and revisions made to the manuscript are summarized in the following.

Reviewer 2 Report

  1. The risk scenarios of intelligent ships are proposed and evaluated by mapping hierarchical holographic modeling (HHM) into risk filtering, risking and management (RFRM). The methods considered here might provide helpful references for relevant fields for evaluation of intelligent ships. However, a separate section “Results and discussion” should be added by separating the content from the research method description for the navigational risk identification and screening of intelligent ships. Moreover, considerably more evaluation results and discussion for this study are suggested to provide more comprehensive evaluation. Hence, major revision is suggested for the manuscript.
  2. Please check and revise the cited references in the text; for example: at lines 55, it could be Wrobel et al.; it could be Utne et al. at line 58, and many others.
  3. At lines 85-87, what are the so-called microscopic models that the authors indicate for the relevant research for intelligent ships?
  4. Please check the English writing in the manuscript. A few typo errors appear, for example, it could be “proposed” at line 61, it could be “established” and “analyzed” at line 66, “safety” at line 68, “performed” at line 76, “technologies” at line 77, “maritime” at line 78, etc. In Fig. 4, is it “status” or “statu”? At line 465, is it “flow” or “flow chart” or other?
  5. There are many mathematical symbols particularly subscripts used in this article. A list of nomenclature is suggested in the revised manuscript.
  6. References are suggested after Delphi method at line 209, after Bayesian method at line 449 and after Bayesian equation at line 450.
  7. Please check the correctness of the flow chart in Fig. 3. For example, if it chooses “No”, then it will only go to “Literature consulting …..”, is the logic correct?
  8. Please rephrase the following expressions, which are confusing and repeated. At line 402-403, “This paper refers to …. for this study”; “This system design lacks are proved in a scene” above the line 406; in the title of Table 13, “Quantitative ranking … navigational risks”.

Author Response

Dear Editors and Reviewers:

Thank you for your letter and for the reviewers’ comments concerning our manuscript entitled “Risk scenarios evaluation for intelligent ships by mapping hierarchical holographic modeling into Risk Filtering, Ranking and Management”. (ID: Sustainability-1591128). The authors are very appreciative of the review made by the reviewers and the valuable comments and recommendations provided for the improvement of the above-referenced manuscript. We have studied comments carefully and have made correction which we hope meet with approval. Revised portion are marked in red in the manuscript. The point to point responses and revisions made to the manuscript are summarized in the attached file.

Round 2

Reviewer 1 Report

The author has made the necessary corrections. It can be published as such.

Reviewer 2 Report

1. The reviewer trusts that the authors have revised their manuscript in response to the comments.